# Which Natural Areas are Preferred for Recreation? An Investigation of the Most Popular Natural Resting Types for Istanbul

**Meryem Hayir-Kanat** [1,*] and **Jürgen Breuste** [2]

1   Faculty of Education, Yildiz Technical University – Istanbul, Davutpasa Campus,
    Esenler-Istanbul 34210, Turkey
2   Department of Geography and Geology, Paris Lodron-University of Salzburg, A5020 Salzburg, Austria;
    juergen.breuste@sbg.ac.at
*   Correspondence: mhayir@yildiz.edu.tr; Tel.: +43-681-8456-1361 or +90-533-643-3193

**Abstract:** This research focuses on people's perceptions and expectations from nature and nature experience and their preferences of nature types. One-on-one face-to-face interviews were conducted with 500 respondents using a paper-pencil survey questionnaire. Our results show that the most frequently cited meaning of nature was fresh air and green space. Overall, the majority of the respondents associated the concept of nature with green areas, coasts, and panoramas. The psychological dimension of nature was also mentioned by about one-third of the respondents. The most beloved part of being in nature reported by the respondents was being away from city life and work. The respondents had strong preferences for being near water, primarily by the seaside. It is concluded that, for many people, nature and biological components of nature help with the recovery from work stress and city hassle.

**Keywords:** perceptions about nature; expected benefits from nature; nature preferences; nature experience; Istanbul

## 1. Introduction

There is rapid urbanization around the world. The largest cities get larger, and green space coverage in cities declines as human population density and urbanization increase [1,2]. Growing evidence shows that urbanization decreases perceived well-being [3] and happiness [4]. The well-being of people living in densely-populated areas is much lower [3,5,6]. These findings lead to concerns about conserving urban green spaces, quality of human life, and the maintenance of human-nature contact [1,7].

Human-nature contact is of profound concern, because such interaction has been proven to have a variety of physical, health, social, and psychological benefits [1,7–18]. Several studies have found that urban green spaces provide ideal settings for physical activity, particularly for walking [15,19–21] and for strengthening social ties [12,14,22,23]. Furthermore, exposure to the natural environment has restorative and mental health benefits [11,13,15–17]. Green areas have been found to reduce the annoyances resulting from city noise [8,9] and to decrease stress levels [11,16,24] and stress-related illnesses [10].

As is the case in many European cities [25], Istanbul has become a high-density city, resulting in the trend of the disappearance of natural green spaces. This research has originated from the raised concerns regarding how and where people meet their needs of daily recreation and nature experience and keep their connection with nature in such a large urban city. It addresses a crucial but relatively

unexplored question on people's perceptions about, preferences of, and expected benefits from nature and nature experience, with the premise that understanding these provides critical information to develop policies and practices focusing on human-nature contact.

## 1.1. Nature, Nature Experience and Expected Benefits

Perception helps us understanding the environment around us [26], and perceiving leads to acting [26,27]. At the same time, a human's ability to act in the surrounding environment influences one's perception. As a result, an environment would look different to different people with varying capabilities, or would even look different to the same person as his or her abilities change [28]. In this research, we explore human's perceptions about nature and nature experiences and their motives for being in nature, assuming that what they perceive would influence their actions; specifically, how and why they interact with nature.

There is not a unified definition of nature. "Nature is that which we observe in perception through the senses" [29]. A more recent and broad definition of nature is:

"physical features and processes of nonhuman origin that people ordinarily can perceive, including the "living nature" of flora and fauna, together with still and running water, qualities of air and weather, and the landscapes that comprise these and show the influence of geological processes." [30].

Bratman et al. [31] differentiate what exists and what people perceive. Specifically, they refer to the first group as nature and define it as "areas containing elements of living systems that include plants and nonhuman animals across a range of scales and degrees of human management, from a small urban park through to relatively 'pristine wilderness'" (*p.* 120), while nature experience is "time spent being physically present within, or viewing from afar, landscapes (or images of these landscapes) that contain elements from the above category" (*p.* 122). From another perspective, the concept of nature may refer to the physical aspects of the earth as well as abstract concepts including a variety of ideas and principles [32]. Ellen [33] describes three dimensions of nature. The first one is "nature as kinds of 'things'" (*p.* 105). The second dimension includes nature as space that is not human ('sea' or 'desert', say, or 'mountains' (*p.* 110). The final dimension is "nature as inner essence (its sensation as an inner essence or vital energy or force outside human control (*p.*111)." Also, the concept of nature means different things in different cultures and at different times [31–33].

People living in urbanized societies use the natural environment as a place to relax and express themselves and to escape the regularity and hustle and bustle of the environment [34]. Exposure to nature may serve as an important facilitator for the recovery of stress resulting from work and city life. According to Sonnentag and Fritz [35], recovery includes physical and psychological detachment from work, relaxation, mastery, and control. Physical and psychological detachment includes not being occupied by any work-related activity or duties, physically or mentally. Relaxation means a person's state of increased positive affect and low activation. Mastery experiences are activities that are engaged in during off-job time, offering challenging experiences and learning opportunities without exhausting one's abilities. Control refers to one's ability to choose which activity to pursue and when and how to pursue it during leisure time. One part of this research includes exploring whether people seek recovery in their nature experience through their perceptions about nature and resting.

Using natural recreational areas may be undertaken to achieve certain psychological and physical goals [36]. Research has identified that escaping the stressors and particularly relieving work stress are important reasons for nature experiences [9,10,37,38]. Several studies have shown that resting and relaxing are the most important motives for park visits [22,23,39,40]. Urbanized natural areas provide feelings of peace and tranquility [41,42]. Some other reasons are to escape from the city hustle and bustle [22,43,44] and stressors [9,10,37], to be with children [22], to observe nature [22,42–45], and to meet other people [22]. Among the least important motives are to do sport, contemplate and meditate, get artistic inspiration [22], and have better physical health [23]. Based on factor analyses, [22] Chiesura classified the motives for nature experience. The first category relates to more within-person benefits such as being on your own, reflecting in peace, and developing artistic and cognitive skills.

The second one is more about interpersonal and social advantages such as socialization and other pleasant activities. Weber and Anderson [46] identified that the common reasons that people visit natural area are to enjoy nature, escape personal/social pressures, escape physical pressures, and enjoy the outdoor climate. Schipperijn et al. [47] examined reasons to visit green spaces from a national sample in Denmark. They found that the most frequently reported reason was to enjoy the weather and get fresh air, followed by the desire to reduce stress and relax. Exercising, doing something together with family and friends, and observing the flora and fauna were less frequently expressed as reasons. Some of the aforementioned reasons or motivations to use nature or nature elements may be conceptualized around "leisure". Scholars have little consensus about the definition of leisure. In addition, not only the meaning of leisure may vary from culture to culture [48,49], existence of leisure-related lexicons across cultures also vary [50]. In Turkish language, resting is one of the lexicon used for leisure. Resting is also used for relaxing, sleeping, recreation etc., which may overlap with the meaning of various terms related to leisure concept. Culturally, resting can be defined as a "temporary cessation or break from a physical or mental activity or work" or "being engaged in an activity such as sitting, or drinking a cup of coffee or going for a vacation for some time". It can be short or long, passive or active, solitary or gregarious. In this study, however, we investigated the meaning of resting to understand whether individuals living in Istanbul associate resting with nature and consider nature or any form of nature elements as a venue to rest defined culturally.

### 1.2. The Present Study

This research focuses on human-nature contact, with an emphasis on self-reported perceptions, expectations, and preferences related to nature and nature experience for people in Istanbul. Specifically, the following research questions are addressed: (1) What does nature and resting (in relevance to nature) mean to people? (2) What does nature offering, and to what extent do people prefer to spend their time in nature? and (3) What are the benefits people expect from nature areas? The findings of this research would act as a means to improve the quality of peoples' lives, to conserve nature, and preserve and improve natural areas to maintain human-nature contact.

Most research on human-nature interaction covers cities in Europe, Australia, and North America, while cities with the most pronounced urban growth elsewhere are underrepresented [51,52]. Also, research has well-established that peoples' perceptions about nature change across countries, time, culture, and individuals [31–33]. Clearly, there is a need for more research focusing on perceptions of people living in different regions of the world. In this sense, this research helps to fill an important gap because it is likely that people living in Istanbul, representing Turkey, might have perceptions about nature different from the perceptions of people living in other parts of the world.

Also, research on people's nature experiences mostly comes from inland cities or towns and includes nature types such as forests, lakes, gardens, urban green spaces, and parks [1,53,54]. There is little empirical research drawn from coastal areas focusing on the perceptions and nature experiences of people living there. Thus, this research contributes to our understanding of nature preferences, experiences, and perceptions of people living in an urbanized city located in a coastal area.

## 2. Materials and Methods

### 2.1. Study Site

Istanbul is a coastal city, located in the northeastern part of Turkey, having coastal sides on the Marmara Sea, Black Sea, Bosphorus, and lakes, totaling about 927 km of coastal length. It is situated in an area of 5712 km$^2$, half of which is a forest (2671.98 km$^2$). It has unique characteristics both geographically and demographically. It lies on both the Kocaeli peninsula on the Asian continent and the Çatalca peninsula on the European continent, with the Bosphorus in-between, bridging both continents. It is also the most populated city of Europe and Turkey, with a growing population of

15,029,231 inhabitants, as of 2017 (See Figure 1 below [55]). The city is settled along the east-west axis of the Marmara Sea coast on both peninsulas and the Bosphorus [56].

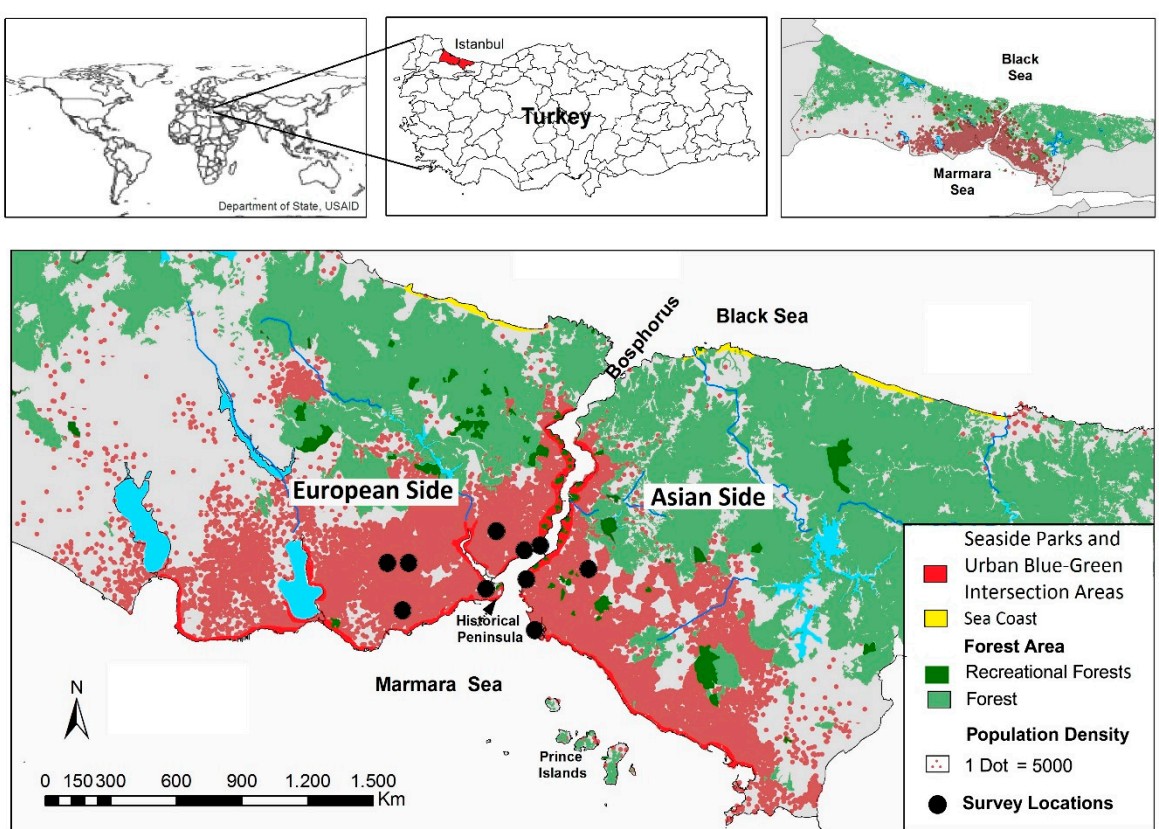

**Figure 1.** Location of Study Area and distribution of nature types for resting in Istanbul.

There are 39 districts in Istanbul with varying social, economic, and demographic characteristics of their dwellers. A group of researchers [57] developed a categorization to describe such characteristics of the districts based on a human development index (HDI), using seven areas of development (governance and transparency, social coverage, economic status, education, health, social life index, the municipal's environmental performance, and transportation). They came up with four groups: Very high HDI, high HDI, middle HDI, and low HDI. Their categorization showed that 16 districts of Istanbul fall into the very high HDI group, 14 districts fall into the high HDI group, and nine districts fall into the middle HDI group. There was no district in the low HDI group.

*2.2. Respondents*

The participants of this study consisted of 500 individuals who volunteered when they were invited to participate in the study and to complete the interview. Of the 500 individuals who were administered the survey questionnaire, 52% were female and 48% were male. 9.4% were 19 or younger, 36.1% were aged 20–24, 12% were aged 25–29, 9.8% were aged 30–34, 10.6% were aged 30–39, 9.8% were aged 40–49, 8.8% were aged 50–59, and 3.2% were 60 or older. Gender distribution by age groups appeared to be similar.

The Occupational status of the respondents was as follows: 9.6% were state employees, 11.8% were workers, 9.4% were self-employed, 7.8% were retired, and 14.2% were unemployed. About 47% of the respondents were students (46%). Of those, 72% were 20–24 years of age and 16% were 19 or younger.

Approximately 83.2% (f = 416) of the respondents were recruited through the interviews at the squares on the European side and 16.8% (f = 84) were recruited through the interviews at the squares on the Asian Side. Also, the participants consisted of those living in one of the 32 districts out of the 39.

Two-thirds of the respondents reported living in a district with very high HDI, about 25% in a district with high HDI, and 9% in a district with middle HDI.

*2.3. Data Collection Sites*

The data was collected at 10 downtown squares out of 65 located around Istanbul. Out of 10 squares (See Figure 1), three squares were located on the Asian side (squares at Kadıköy, Üsküdar and Ümraniye) and seven squares were located on the European side (squares at Eminönü, Taksim, Beşiktaş, Ortaköy, Esenler, Bağcılar, and Bakırköy). These squares were chosen because they are located close to the city center, are identified as being more significant squares than others (Çakılcıoğlu, Reyhan, and Kurt, 2010), and are popular because of their historical, touristic, shopping, entertainment, and sightseeing locations. Thus, these squares attract many individuals from diverse socio-economic backgrounds and with different interests around Istanbul, which, in turn, allows researchers to reach a sample as diverse as possible.

The data were collected in November 2016. People passing by the researchers at the squares were approached at random and asked if they would agree to participate in the survey. Of those who agreed, only residents were interviewed. Face-to-face interviews were conducted with those who volunteered (34%) to participate in the research. Interviewers asked the questions to the respondents one-on-one and completed the paper-pencil survey questionnaire based on their answers.

2.3.1. Survey Instrument

The survey instrument consisted of three parts. The first part included items addressing the self-reported demographics of the respondents, including respondents' age, gender, occupation, and in which districts of Istanbul they lived.

The second part was about the meaning of and general attitudes towards nature. In this section, two questions were asked. The second part was concerned with the respondents understanding of nature and included two open-ended questions: "What does nature mean to you?" and "What does resting mean to you?"

The third component of the survey included questions aimed at getting insights into where and why Istanbulites go to nature and which needs they expect to be fulfilled when they get to the nature area. Respondents were first asked which activity they preferred among the following; "spending time at the coast", "walking in the forest", "walking in the city park", "doing sports at a sports facility", or "other". Then, to find out the degree to which the respondents would like to spend their time in a specific nature type, they were asked to rate each area using the Likert scale: "I would want a lot", "I would want" "I would want a little", and "I would not want". They were provided with four photos, taken during the daytime in the first days of November 2018, representing the main natural area types in Istanbul; nature area on the seaside (Figure 2A); nature area near a lake (Figure 2B), forest area (Figure 2C), and an urban park (Figure 2D).

The last question was "What do you like most about being in nature?" The response categories included "escaping from the city noise", "recovering from work stress", "spending time with family", "teaching children about the natural environment", "benefit of it on healthy life", and "other". Respondents were allowed to give more than one answer for both questions.

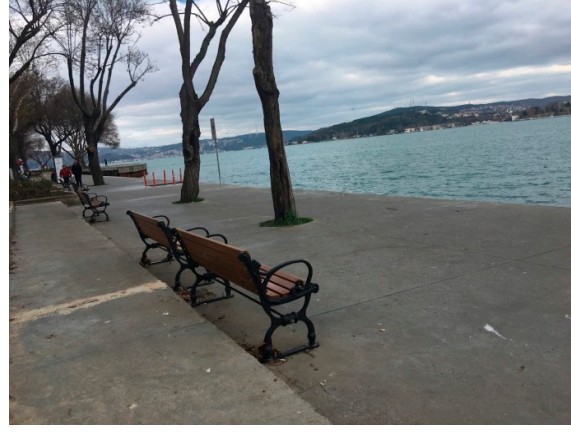

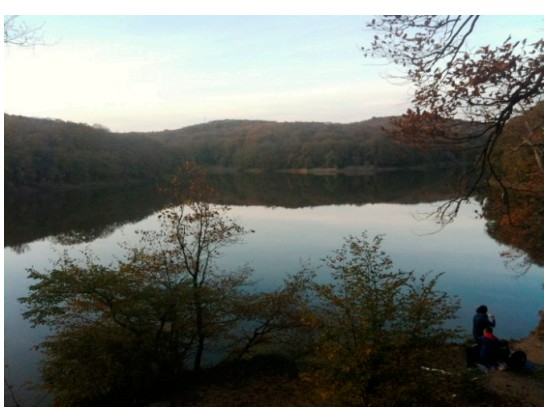

(**A**) Nature area on the seaside (Bosphorus)    (**B**) Nature area near a lake (in Belgrade Forest)

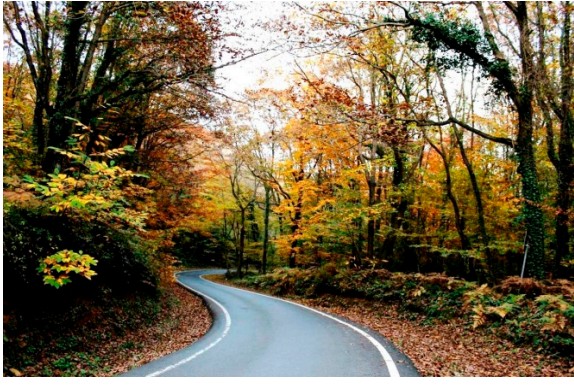

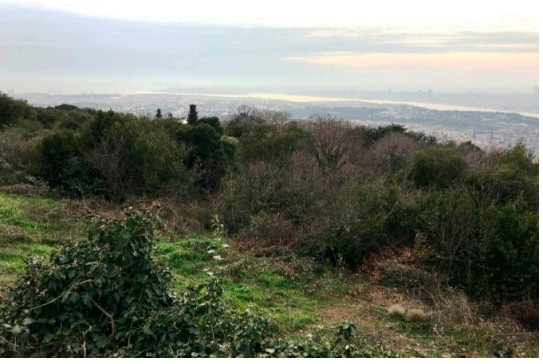

(**C**) Forest (Belgrade Forest)    (**D**) Urban park (Camlica Hill)

**Figure 2.** Photos shown to respondents to rate which nature area they would like to spend their time at (Photos, Meryem Hayir–Kanat).

### 2.3.2. Data Analysis

Respondents were allowed to give more than one answer to all questions. Responses to open-ended questions included both behavioral patterns and abstract terms, which resulted in a data set required a careful interpretation. However, because of the data collection method and interview techniques applied, we were not able to verify or clarify participants' responses. In an attempt to minimize possible errors or flaws due to such issues, we used categorizations and terms used in the literature [1–3], during coding. We believe that the formed categories are reasonable, however, they could be further evaluated by future surveys.

Statements given to open-ended questions were first grouped and coded based on the similarities of their meanings. Then, these codes were regrouped and main categories were obtained. The coding procedure was conducted with a group of three trained graduate students. Discussions were held for each response until 100% agreement was reached. Then, frequencies and percentages were calculated over the number of responses for each response category.

## 3. Results

### 3.1. Meaning of and General Attitudes Towards Nature

In this part, to understand what resting and nature mean to respondents, we asked respondents the questions "What does nature mean to you?" and "What does resting mean to you?" Table 1 provides categories formed based on respondents' answers.

**Table 1.** Meaning of Resting and Nature.

| Meaning of Resting (%) | | Meaning of Nature (%) | |
|---|---|---|---|
| Sleeping | 16.9 | Area with abundant oxygen/green areas | 45.6 |
| Not working or having any responsibility | 14.2 | Sea-lake coast | 13.9 |
| Spending time with family | 8.9 | Peaceful or relaxing area | 12.6 |
| Being in a natural environment | 7.2 | The true source of life | 8.8 |
| Having peace of mind | 6.6 | Area away from city life | 5.7 |
| Relaxing | 6.2 | Bird sounds | 4.6 |
| Being away from noise | 5.5 | Area with lots of bugs and beetles | 2.4 |
| Having peace of mind at home | 4.9 | Silence | 2.2 |
| Walking/Doing Sports | 4.5 | Landscape/View | 1.3 |
| Reading | 3.9 | Recreational area | 1.3 |
| Listening to Music | 3.9 | Others | 1.6 |
| Watching Movie/TV | 3.7 | | |
| Meeting friends | 3.4 | | |
| Walking | 3.4 | | |
| Sitting by the coast | 2.9 | | |
| Being alone | 2.1 | | |
| Having tea/coffee | 1.8 | | |
| | 100 | | 100 |

Regarding the meaning of resting, about one-third of the responses involved engagement in some type of lethargic activity such as sleeping, not working, or not having any responsibility. Another 25.5% of the responses associated resting with activities that seek peace and reflection, such as being away from noise, being alone, relaxing, or having peace of mind. About 12% of responses associated resting with spending time with family and friends and 10% involved doing activities in nature such as being in the natural environment and sitting by the coast. However, it is highly likely that some of the activities listed in Table 1, such as having tea or coffee or meeting friends and sitting by the coast, may be done simultaneously and some activities, such as being away from noise, walking, doing sports, or being with family or friends, may be done in nature.

For about 45.6% of the respondents, nature was associated with fresh air and green areas. Sea/lake coasts were the only concrete named landscapes understood as nature from 13.9% of the respondents. Approximately 12.6% thought the meaning of nature was being in peaceful and relaxing places. About 1.8% associated nature with other things. It seemed that for the majority of respondents, nature was associated with a physical component of nature such as a green nature area, fresh air, etc., while about one-fourth (those finding the meaning as "the true source of life", "peaceful and relaxing places", and "silence") associated it with more psychological components.

*3.2. Natural Recreation Area Preferences and Expected Benefits from Using Nature Areas*

3.2.1. Activity Type Preferences

When respondents were asked which activity they preferred, in general, among four types of activities (depicted in Figure 3), 90% reported that they did an activity in the natural environment or an open space. The most preferred activity was spending time at the coast, with 42.9%, followed by walking in the forest then by walking in a city park.

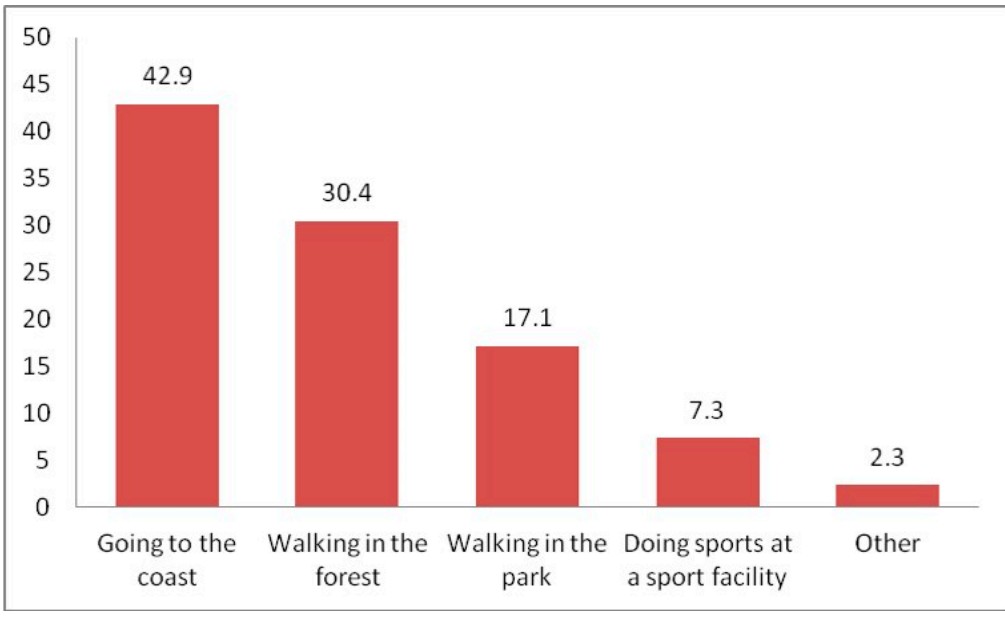

**Figure 3.** Activity preferences (in percentages).

### 3.2.2. Nature Type Preferences

To find out which natural recreational area the respondents would prefer, the respondents were provided with four photos, representing the main natural area types in Istanbul (see Figure 2A–D above). They were asked to rate their desire to be at the specific area on a four-point Likert-like scale from "I would want a lot" to "I would not want". The percentages are provided in Table 2.

**Table 2.** Nature type preferences.

|  | I would want a lot (%) | I would want (%) | I would want a little (%) | I would not want (%) |
| --- | --- | --- | --- | --- |
| Nature area on the seaside | 69.1 | 25.5 | 4.4 | 1.0 |
| Nature area near a lake | 37.9 | 42.4 | 13.5 | 6.2 |
| Forest | 44.2 | 36.1 | 15.8 | 3.9 |
| Urban Park | 32.8 | 30.5 | 22.1 | 14.6 |

The percentages show that people had strong preferences for nature areas on the seaside. About 95% of the respondents expressed that they would want to be by the seaside. Approximately one-third of the respondents said that they would "want a little" or would "not want at all" to be in an urban park. Degrees of preferences were similar for nature areas near a lake or forest.

### 3.2.3. Expected Benefits from Nature Areas

Figure 4 shows what respondents liked most about being in nature in percentages. At 52.5%, the most liked part of being in nature was to be away from city noise. This was followed by relieving work stress, having family time in the natural environment, and having a healthy lifestyle. Teaching children about nature was the last priority.

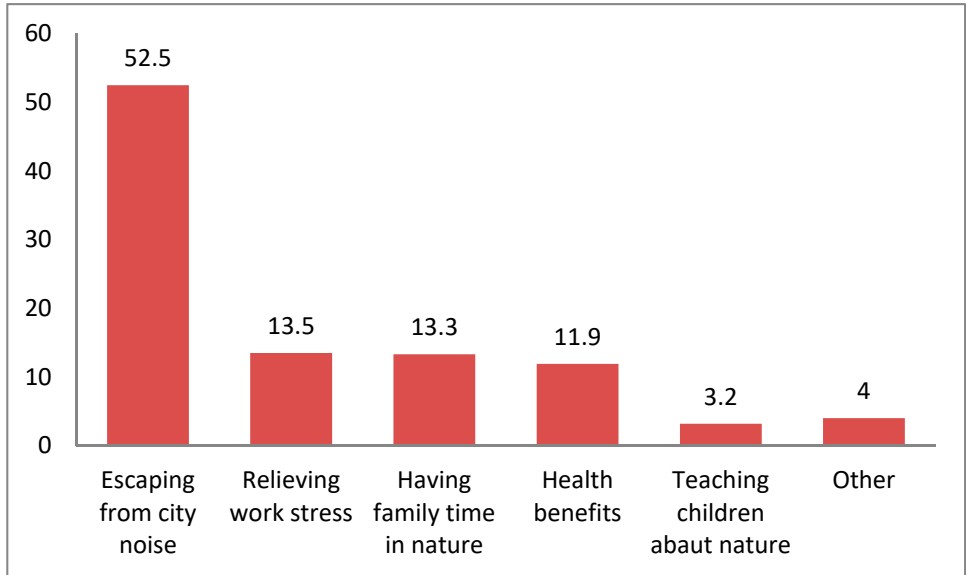

**Figure 4.** Expected benefits of being in nature (in percentages).

Next, in order to explore whether expected benefits were associated with preferred activity types, a crosstabulation of these two variables were run. Table 3 shows the percentages of respondents who use nature for different expectations and their activity type preferences. It appears that 50% of the respondents who expected to escape from city noise, over 37% who expected to relieve work stress, and 44.1% of the respondents who expected health benefits preferred to go to the coast as an activity type. On the other hand, 81% of the respondents who expected to teach children about nature preferred to walk in an urban city park (50%) or in the forest (31.3%)

**Table 3.** Activity type crosstabulated against expected benefits.

|  | Escaping From City Noise | Relieving Work Stress | Having Family Time in Nature | Teaching Children About Nature | Health Benefits | Other |
|---|---|---|---|---|---|---|
| Walking in the Forest | 29.2 | 26.9 | 36.4 | 31.3 | 32.2 | 33.3 |
| Doing Sports at a Sport Facility | 5.8 | 11.9 | 1.5 | 6.3 | 13.6 | 11.1 |
| Walking in an Urban Park | 11.9 | 22.4 | 31.8 | 50.0 | 10.2 | 14.8 |
| Going to the Coast | 50.4 | 37.3 | 28.8 | 12.5 | 44.1 | 33.3 |
| Others | 2.7 | 1.5 | 1.5 | 0.0 | 0.0 | 7.4 |

## 4. Discussion

This research has focused on people's perceptions about resting and nature, the expected benefits from nature and nature experience, and their preferences of nature and resting activity types. Key findings of the research are as follows:

For about 73% of the participants, nature was associated with a sound, a scene, a place, a state, or an area related to green or nature spaces.

For only 10%, resting was associated with doing activities in nature (specified as being in nature or sitting by the coast), while the majority perceived resting as being away from daily routines or passive activities such as sleeping and not working.

Among four nature types—sea side coast, near lake area, forest, and urban parks—sea side coast was the most frequently desired place where respondents wanted to be. Similarly, the most frequently preferred activity was spending time at the coast.

The most frequently cited benefit people expected from nature was escaping from city noise.

### 4.1. Meaning of Resting and Nature and Expected Benefits from Nature

Over half of the respondents associated the concept of resting with "being less active physically or mentally" and "being away from daily hassles". Nature experience did not seem to be the most preferred activity for resting. A small percent of respondents (10%) associated leisure with being in nature, without specifying what they do there; whether they reflect, enjoy the landscape, socialize, or something else. Thus, we do not necessarily know which aspect of nature helps them rest for this small percentage.

The most cited meaning of nature was fresh air and green space. This is consistent with a study by Schipperijn et al. [47]. Also, the majority of the respondents associated the concept of nature with green areas, coasts, and landscapes. The psychological dimension of nature was also mentioned by about one-third of the respondents. People have an innate love for the natural world [58], but the degree of importance they place upon nature, the way they benefit from and interact with it, and how they perceive it vary across cultures, even across individuals [31–33]. Consistent with the literature, the present study provides evidence that nature means different things to different respondents. Nonetheless, the respondents' statements associated with nature can be classified under two general categories. The first one, which respondents featured prominently, was nature as a space [33], including green areas, coasts, and landscapes (about 60%), with ecological components such as birds, bugs, and beetles (about 7%). These meanings associated with nature align with the definition made by Bratman and colleagues [31]. On the other side, about 30% of the respondents mentioned the restorative dimensions of nature, specified by Sonnentag and Fritz [35], such as nature being "peaceful", "relaxing", "silent", and "away from city life". Relevant to the latter, being away from city noise and relieving work stress was the most loved part of being in nature for more than two-thirds of the respondents. This shows that respondents were more often motivated for nature experience by restorative reasons. In our study, "to escape city noise", was the most frequently mentioned motive by over half of the respondents. This was followed by "to relieve work stress". Remember that half of the respondents associated leisure with stillness and passive activities such as not working/having no responsibilities, sleeping, being peaceful, body equilibrium, being away from noise, or staying at home. These two consistent findings, along with findings of other studies conducted in urban cities, e.g., [22], can be explained by the fact that urban cities, particularly those like Istanbul, are characterized by overcrowded population, noise, and a hectic rhythm of life. Also, for many people, nature and the biological components of nature help in the recovery from work stress and city hassle. Evidently, what people associate with nature or nature experience influences how people interact with nature and vice versa. As the importance of green areas in reducing the annoyances resulting from city noise is evidenced in the literature [8,9,37], providing urban green spaces is critical to maintaining well-being, particularly the mental health, of individuals.

While people seek an array of social, physical, and psychological benefits through nature experience [21], spiritual benefits may also be included. As specified by our respondents, nature areas provide opportunities for being away from work, responsibilities, people, or noise, and for being peaceful, having body equilibrium, and being on your own. Here, based on our respondent's self-report, aligned with the literature [41,42], experiences at urban green spaces and natural recreational areas provide a feeling of peace and tranquility. Taken together, it appears that for 80% of the respondents, meeting their psychological needs through these three motives is the most important benefit of nature experiences. Particularly, accessible urban nature away from city centers, noise, and traffic provides a "safe haven" for people living in big urban cities.

"To have family time" constitutes another important reason to experience nature. About 13% of the respondents spent their time in natural recreational areas socializing by doing things such as meeting with their friends and family members. In that sense, our findings, along with the existing literature, e.g. [22], provide evidence that nature helps increase family ties and fulfill social functions.

Applying Chiesura's terminology [22] to what respondents attribute to nature, it is possible to conclude that some definitions given by our respondents included amenity dimension reflecting

divergence from routine life such as "being away from city life", "listening to bird sounds", "observing natural beetle areas", "coasts", and "green areas", spirituality dimension such as "true source of life"; and restorative dimensions, reflecting re-creation of body-mind equilibrium, such as "peaceful and relaxing places". Their search for a restorative function of nature was also reflected in their responses to what motivated them to go to nature as they looked to relieve work stress. The restorative/positive effect of nature on stress has also been identified [10]. Although amenity dimension seems to be more frequently reflected in respondents' definitions, as interpreted by Chiesura [22], such restorative and spiritual perceptions are the reflection of ecological and environmental awareness.

## 4.2. Nature Type and Activity Preferences

The number one preferred natural activity type by the respondents was spending time by the coast, followed by walking in the forest, then walking in a city park; all are in open spaces. Doing sport was preferred by a very small proportion of the respondents. Also, respondents had strong preferences for being near water, primarily by the seaside, to escape from city noise, relive work stress, and health benefits. These findings show that the favorite environments of people in Istanbul are primarily coastal sites. The beauty of landscape appears particularly to be an important element that explains people's love of these sites. Several other studies originating in Istanbul have also reported that the beauty of nature or naturalness of the recreational area seems to be the most selective criterion for people visiting natural recreational areas in and around Istanbul [59,60]. These findings are consistent with more global research, which has shown that coastal areas are recognized for their attractive scenery and recreational activities [61].

A nature area by a lake and forest appear to be preferred similarly, while urban areas are preferred less. There is particularly less interest in experiencing forests. This could be explained by the fact that forest areas are far from the city center and have limited public transportation opportunities and walking infrastructure. Also, when the photos are looked at carefully, the forest and near-lake photos include a scene with a lot of trees and greenness, whereas the photo of the urban park seems to be far from the sea only with a far-visual contact and with a little vegetation. Also, two photos have humans and benches, etc., in the settings, which may facilitate the respondents to envision the settings easily and may lead to a preference for these two, whereas the other two do not have such components. Thus, there may be a photo effect in that the quality of the photo and the bias selecting the scene [62] may influence people's perceptions about those areas. This presents a limitation to the study. On the other hand, respondent's nature and activity preferences to the questions without a picture and with a picture show similarity, which may be a relief.

Another possible limitation of this study could be the questionnaire. In the questionnaire, we asked for individuals' natural area preferences without presenting clear definitions of these different types. We first prompted the nature types only verbally for a question, then provided a general picture for another. Being physically present in an area involves experiencing the area with multiple senses; seeing a photo of an area has mainly a visual prompt [63], and asking verbally involves only a verbal prompt. People probably express a different level of desire to be at an area and evaluate their experience and perceptions differently when they are only verbally asked, shown a picture, or are in that area physically. Thus, we do not exactly know what they meant by the coast; for example, whether they meant an area with a view of the sea (visual contact), or parks or another area with swimming and fishing options.

Respondents' demographic characteristics probably influence the preferences, nature experiences, and perceptions. For example, Schipperijn et al. [47] found differences in peoples' main reasons to visit green spaces by gender and age. They found that older respondents were more likely to visit green spaces to follow the seasons and observe flora and fauna, while younger respondents' main reasons were to enjoy the weather and get fresh air. Also, retired people less often sought stress relief from the nature experience. Considering the demographic composition of our respondents, about half of whom were aged between 20 and 29 and were students, our findings are consistent with their

study. As the sampling response bias on the demographic variables of age and occupation status presents a possible limitation of this study, whether respondents' demographic characteristics would influence their preferences, nature experiences, or perceptions remain a question. Future research should consider these limitations.

Despite the encouraging results obtained, some results need to be more closely examined in the future. For example, what people expect from a natural environment that would serve to recover from or reduce the stress from an urbanized and densely populated city life are topics for further study. When researching these topics, socio-demographic characteristics of individuals deserve special attention.

**Author Contributions:** Conceptualization, M.H.-K. and J.B.; data curation, M.H.-K.; formal analysis M.H.-K. and J.B.; investigation, M.H.-K. and J.B.; methodology, M.H.-K. and J.B.; project administration, M.H.-K. and J.B.; supervision M.H.-K. and J.B.; visualization M.H.; writing-original draft, M.H.-K.; writing-review and editing, M.H.-K. and J.B.

**Funding:** No external funding.

**Conflicts of Interest:** The authors declare no conflict of interest.

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
