# Peer review of "Which Natural Areas are Preferred for Recreation? An Investigation of the Most Popular Natural Resting Types for Istanbul"

_sustainability, doi:10.3390/su11236773_

Round 1

Reviewer 1 Report

Thank you for this opportunity to review your work.  This is an interesting study, and the constructs studied are complex and challenging to undertake.

Section 1.2, line 112; the authors indicated part of the significance of this study is that it is in a developing country, and much of research has focused on developed countries; also the phrasing of Istanbul representing Turkey is used; and thus the study of Istanbul people is used as an indication of nature experience/preference/perception in a developing country.  I don't know enough about Istanbul nor Turkey, but as a larger city, I would think that there is quite a variation as to perceptions/preferences etc from those living in Istanbul to more rural regions.  Because so, I would caution against the "jump" from Istanbul to "developing countries".  I think the point that this changes across time, place, culture etc warrants this study v. that the research was conducted in Istanbul and thus a developing country

Section 2.2.1 Respondents  needs editing; seems incomplete.  Also, how was the sample obtained? Also curious as to the oldest age bracket being 40 and older.  Was that because few respondents or no respondents in an older age category?  It seems there is literature as to how recreation behaviors/motivations etc changes with life stage, and to not have respondents in the older age categories potentially could be a limitation and/or misleading, as results may have been different if older participants.  

Section 2.2.2 data collection site: Was there a survey item asking if resident v. tourist?  It seems this is an area where tourists go, and thus potentially tourists v. residents could have been in the sample of respondents, and tourists likely would or could have different responses to these items than residents. Based on the info in the next section, it doesn't appear this was asked.  Thus, to have respondents and to not know if they are from Istanbul seems potentially quite problematic.  Line 147 - please say more about the respondents being in the 3 groups of human development index - does that mean three income categories?  I think that would be useful information when describing the participants in the prior section.  Look at these two sections 2.2.1 and 2.2.2 for organization, as some of the information about the participants is reported in the data collection 2.2.2 site that would be more helpful in the prior section toward understanding the sample.  Also tell more about the "randomly asked" as people walked by - was this truly random?  Can a researcher truly randomly ask without a plan to be intentionally random in our sampling, or instead was this a convenience/volunteer type of sampling - whomever the research was able to approach and ask v. some systematic random sampling

Section 2.2.3 Survey Instrument - the authors reference a qualitative method and use it interchangeably with open-ended questions; please adjust phrasing as open-ended questions on a survey doesn't make this section "qualitative methods", particularly as the way the data is used and interpreted is still within the overarching quantitative paradigm.  Also more detail is needed regarding describing this second section of the instrument; it is stated two questions asked to get at meaning and attitudes; each of these are really quite "big" and complex.  Specifically what was asked and how was the phrasing?  This is important for the reader to know how this might have influenced responses.  Regarding activity choices, I am wondering how the researchers arrived at the set of 5 possible responses.  While "other" provides a way to account for the relatively limited choices, could respondents indicate what that "other" was, or was it just the close-ended other?  Seems the set of choices limits what respondents might choose had there been a broader set of choices...  I also wonder about the 4 choices/4 photos.  One has a human and the others do not - and we know from this body of literature that the degree to which people can envision humans within the setting influences perceptions and preferences, so that adds some "noise" and/or bias (adds another variable) into the mix that complicates findings; similarly we know lighting/seasons/ etc. influence preferences, as does scale, and degree of human maintenance - that one shows a bench and another a trail v. two that don't (that may cause some to choose those two over the ones without it, that may have nothing to do with the landscape setting or particular place); where respondents also give the location name?  That matters as well (preferences being influence by novelty and familiarity of setting).  While it is too late to go back and re-design, very important that this complexity and also the limitations of approach used are acknowledged and discussed!  Same could be said/asked regarding the choices for why they liked spending there - why this set and not a broader set of reasons/motivations (which the literature has many more) - how was it decided to narrow to these?  For other, could people state what the "other" was or do they just have to check "other"?  I think you may be oversimplfying and narrowing reasons.  Also, what if they were motivated by the social opportunity but not "family" but "friends" for example - would they have chosen that response?  This is striking me as over simplified.

Results - the two terms recreation and resting appear to be used interchangeably, when they are not (leisure as a term would be more similar to resting); consistency in phrasing/stating them is needed; also the last two lines of the first section of results states: All categories, except for "the true source of life", "peaceful and relaxing places", and "silence"  indicates a direct link between the meaning of nature and a green, recreation or nature area.  Is this "link" conceptually or statistically?  More explanation is needed as to how this can be said

Results for the next section seem problematic (and this relates back to my prior comments regarding the survey).  The subheaading is natural area preferences, but then the results are phrased as preference for activity (not natural area setting); so there are two things getting mixed together - setting preference and activity preference.  Even the response choices are problematic as one is a place (coastland but openended as to what one would or could do) and the other three signal what activity and a setting (walking, playing a sport), so there are two variables v. one.  For example, what if someone would have chosen forest, if they could have said spending time with family, or watching birds, v. walking on a trail.  So for these reasons, the results get difficult to interpret (one can summarize the findings, but interpreting what they mean becomes the problem).  Similarly, the desired benefit of stress reduction and preference for that makes me wonder if the reason the coast/seaside was preferred was because the verb was "going" and what they did could be something like just sitting/being/relaxing, whereas if they chose forest, they were also having to choose hiking, and maybe that wouldn't accomplish the stress reduction for them in the way another form of recreation might.  

While I appreciate that there is an inclusion of limitations from the survey, I think the manuscript would be improved with a more careful effort to interpret the findings in light of the limitations (can the results be interpreted as they have been, in light of these limitations). I think the results and discussion tend toward an oversimplification of multiple constructs, each of which is very complex and nuanced.

Author Response

Dear Editors

On behalf of my co-authors, we thank you very much for giving us an opportunity to revise our manuscript, we appreciate editor and reviewers very much for their positive and constructive comments and suggestions on our manuscript.

We would like to express our great appreciation to you and reviewers for comments on our paper. Looking forward to hearing from you. Here below is our description on revision according to the reviewer one’s comments. Please see the attachment.

Reviewer 2 Report

This manuscript presents study that investigate Istanbul citizens’ perceptions about nature and their preferences on natural types. This is a nice study. However, there are some points that need to be improved before the ms being considered for publication.

(a)        Sampling framework: we need to know how close is the structure of the sample (sex ratio, age structure, employment etc) to that of the population of the city. I mention, for example, that half of the sample was students. In addition, we need to know if all respondents was resident or there also were some visitors.

(b)        Lines 145-149. Authors need to give more details why this study is representative of the views of the Istanbul population. What is the human development index? Please take into account that there will be readers outside from Turkey.

(c)        In the results presentation it would be nice to see some statistics. E.g. how people preferences for natural types differed relative to gender, age or employment?

(d)       Lines 232-248. This part is not belonging to results and has to move to the conclusion section.

Author Response

(The authors gave the same response as above.)

Round 2

Reviewer 1 Report

Overall I am satisfied with the revisions made.

One area to consider further revision is the new text (in red) on p 7, lines 218-222.  I think I somewhat understand what you are trying to say, but because I also read your response/cover letter. I think this needs to be re-phrased toward having it be understood by readers.  

Also where the term leisure has replaced the term resting (and recreation?), I appreciate the revision toward consistency, but I wanted to check if this switch matches what actually was asked to participants (if they were asked about recreation for example, then it is "too late" to switch the term to something else in the presentation of the results).  If the word now used doesn't match what the participants responded to in the interview question, then there needs to be a different "solution" or response to the concern raised.

In general, I believe the requested revisions have been sufficiently attended to.

Author Response

Response 1:

we have revised the text. Plese see the lines 228-234 in the manuscript

Response 2:

we have given careful considerations about the terms leisure and resting and decided that we asked respondents resting and meant resting. To prevent and confusion or "lost in translation issue", we gave the operational definition of resting by clarifying the meaning of resting culturally and lexicons used for it in the manuscript. Please see lines 108-120

Thank you

Reviewer 2 Report

Dear Editor,
authors responded to my comments successfully. It is important that they recognized some limitations of their work, especially in the sampling frame of their study.

Author Response

Thank you
